**Data Availability Statement:** The data underlying this study are available on Dryad (https://doi.org/10.5068/D11M3Q).

# Identification and selection of optimal reference genes for qPCR-based gene expression analysis in *Fucus distichus* under various abiotic stresses

**Marina Linardić**[1,2]*, **Siobhan A. Braybrook**[1,2,3]*

**1** Department of Molecular, Cell and Developmental Biology, University of California Los Angeles, Los Angeles, California, United States of America, **2** Department of Energy Institute of Genomics and Proteomics, University of California Los Angeles, Los Angeles, California, United States of America, **3** Molecular Biology Institute, University of Los Angeles, Los Angeles, California, United States of America

* marina.linardic@cantab.net (ML); siobhanb@ucla.edu (SAB)

## Abstract

Quantitative gene expression analysis is an important tool in the scientist's belt. The identification of evenly expressed reference genes is necessary for accurate quantitative gene expression analysis, whether by traditional RT-PCR (reverse-transcription polymerase chain reaction) or by qRT-PCR (quantitative real-time PCR; qPCR). In the Stramenopiles (the major line of eukaryotes that includes brown algae) there is a noted lack of known reference genes for such studies, largely due to the absence of available molecular tools. Here we present a set of nine reference genes (*Elongation Factor 1 alpha (EF1A)*, *Elongation Factor 2 alpha (EF2A)*, *Elongation Factor 1 beta (EF1B)*, *14-3-3 Protein*, *Ubiquitin Conjugating Enzyme (UBCE2)*, *Glyceraldehyde-3-phosphate Dehydrogenase (GAPDH)*, *Actin Related Protein Complex (ARP2/3)*, *Ribosomal Protein (40s; S23)*, *and Actin*) for the brown alga *Fucus distichus*. These reference genes were tested on adult sporophytes across six abiotic stress conditions (desiccation, light and temperature modification, hormone addition, pollutant exposure, nutrient addition, and wounding). Suitability of these genes as reference genes was quantitatively evaluated across conditions using standard methods and the majority of the tested genes were evaluated favorably. However, we show that normalization genes should be chosen on a condition-by-condition basis. We provide a recommendation that at least two reference genes be used per experiment, a list of recommended pairs for the conditions tested here, and a procedure for identifying a suitable set for an experimenter's unique design. With the recent expansion of interest in brown algal biology and accompanied molecular tools development, the variety of experimental conditions tested here makes this study a valuable resource for future work in basic biology and understanding stress responses in the brown algal lineage.

**Funding:** The Braybrook group at UCLA is funded by The Department of Cell, Molecular and Developmental Biology and The College of Life Sciences (S.A.B); this work was majorly supported by the U.S. Department of Energy Office of Science, Office of Biological and Environmental Research program under Award Number DE-FC02-02ER63421 and the US Department of Energy (Biological and Environmental Research (BER), the Biological Systems Science Division (BSSD); M.L, S.A.B).

**Competing interests:** The authors have declared that no competing interests exist.

## Introduction

Brown algae represent one of the five major lineages that have developed a multicellular body organization independently from other species (with red algae, plants/green algae, fungi, and metazoans as the other four) [1,2]. Among all of the algal groups, the brown algae have the largest diversity in size and morphology, from filamentous to 'complex' thalli (bodies) [1]. In addition to their complex morphology, brown algal life cycles are similarly diverse, exhibiting a broad range of variation between the gametophyte and sporophyte generations [3]. As the main inhabitants of the intertidal zone, brown algae have to deal with extreme conditions; tidal cycles expose them to desiccation and osmotic shock, evaporation, and heat shock daily. Furthermore, anthropogenic impact (e.g. pollution) serves as an additional source of abiotic stress in these already challenging environments. How brown algae survive, and thrive, in these environments remains unknown. Taking into account their divergent evolutionary history [4], importance as an ecological and economical resource [5,6], and their astonishing ecology [7], the brown algae represent a unique and important group to explore. However, they have been extremely understudied on a molecular level. It is only within the past decades that scientists have begun to explore their rich molecular biology.

Gene expression analyses are necessary to understand how brown algal genetic networks dynamically change upon stimulus to regulate their responses to stressful life conditions. In recent years, microarrays and next-generation sequencing (NGS) technologies have arisen as the most used methods for quantification of gene expression on a global level. In spite of these advancements, qPCR remains one of the simplest and accessible methods for studies of small gene numbers. It also serves as a confirmational tool for NGS-derived results. qPCR is most widely used for fast and reliable quantification of mRNA steady-state levels because of its high sensitivity, accuracy, specificity, reproducibility, and low cost [8,9]. However, the accuracy and reliability of qPCR experiments are highly affected by several factors such as RNA integrity, reverse transcription efficiency, and primer efficiency [10]. It is therefore of utmost importance to combat methodologically introduced variation by normalizing to stable reference genes as internal controls. An ideal reference gene should have a similar expression in all tissues and experimental conditions. Choosing reference genes with unstable expression could result in misleading results and inappropriate conclusions regarding target gene expression. However, the expression of commonly used reference genes may vary depending on the life stage of the organism, experimental conditions, as well as tissue source [10]. Therefore, it is unlikely that a single reference gene would exist for all experiments and it is necessary to evaluate the best reference genes for each experiment. To quantitatively evaluate the efficiency and stability of reference genes for qPCR experiments, several statistical algorithms have been developed, such as geNorm [8], NormFinder [11], and BestKeeper [12]. These methods have been used to determine the best reference genes in red, green, and brown algae [13–16], plants [17–22], and metazoans [23–27].

In some brown algae such as *Ectocarpus siliculosus* and *Undaria pinnatifida*, efforts have been made to identify reference genes, with results indicating variable stability of candidate reference genes depending on experimental treatments [13,14]. The brown alga *Fucus* has been an important cell biology model. Experiments in *Fucus* led to several crucial discoveries in cell polarization and asymmetric cell division [28–32]. Recent studies in *Fucus* further show its scientific importance as a cell biology model [33–39], but also as a tool to investigate ecological and physiological effects of abiotic stresses in natural habitats [40–46]. As new molecular methods become available, the potential for *Fucus* to serve as a modern model system for exploring biology in extreme environments becomes tractable. qPCR serves as a basis to address questions of gene expression; it is, therefore, necessary to develop a set of

normalization genes in *Fucus* to enable reproducible quantitative gene analysis by qPCR. In *Fucus*, *Elongation Factor alpha*, *Beta-Actin*, *Tubulin*, and/or a *14-3-3* protein have all been used as reference genes for qPCR [41,42,44,45]. However, there has been no systematic study of the stability and suitability of such reference genes over a wide array of conditions.

In this study, we identified and tested a set of potential normalization genes for *Fucus distichus*. Nine 'housekeeping' genes were selected as candidate reference genes: *Elongation Factor 1 alpha (EF1A)*, *Elongation Factor 2 alpha (EF2A)*, *Elongation Factor 1 beta (EF1B)*, a *14-3-3* gene, *Ubiquitin Conjugating Enzyme (UBCE2)*, *Glyceraldehyde-3-phosphate Dehydrogenase (GAPDH)*, *Actin Related Protein Complex (ARP2/3)*, *Ribosomal protein (40s; S23)*, and *Actin (ACT)*. Their expression was analyzed by qPCR in samples submitted to various stress conditions; salinity, desiccation, pollution, nutrient deprivation, wounding as well as temperature, light, and phytohormone treatment. Three algorithms were used to evaluate the expression stability of the reference genes: geNorm, NormFinder, BestKeeper; a rank aggregation algorithm was employed to reach a consensus between the three. The number of reference genes to use for qPCR-based normalization was also explored. We provide a recommendation of paired reference genes for the conditions tested, alongside a recommended method for suitability assessment in an individual's experiments. To validate our recommendations, the differential expression of *Hsp70* and *Hsp90* genes, encoding stress-responsive heat shock proteins, were examined under salinity stress.

## Results

### Choosing candidate reference genes

To identify the 'housekeeping' gene sequences for potential normalization genes, two approaches were taken. First, housekeeping gene sequences previously reported for *Ectocarpus siliculosus* [13] were aligned against the related species *Fucus serratus* embryo development transcriptome [36] to identify putative homologs. Furthermore, additional sequences were identified directly from the *F. serratus* transcriptome dataset, by identifying highly and constantly expressed genes across the four embryo developmental stages in the study. As such, candidate gene sequences were chosen based on three criteria: 1) a high percentage of alignment to *E. siliqulosus* (for those with *Ectocarpus* sequence matches), 2) high expression values in the *Fucus* embryo transcriptome (normalized expression metric 'transcripts per million transcripts'; 'TPM') and 3) a relatively constant expression across samples in the *Fucus* embryo transcriptome. Nine housekeeping genes were selected that satisfied these criteria: *elongation factor 1 alpha (EF1A)*, *elongation factor 2 alpha (EF2A)*, *elongation factor 1 beta (EF1B)*, a *14-3-3 gene*, *ubiquitin conjugating enzyme (UBCE2)*, *glyceraldehyde-3-phosphate dehydrogenase (GAPDH)*, *actin related protein complex (ARP2/3)*, *ribosomal protein (40s; S23)*, and *actin (ACT)* (Table 1, sequences in S1 Fig). Primer sets were designed for all nine genes and their efficiencies and melting temperatures determined by qPCR (Table 1, S1 Fig; See Materials and Methods). The efficiency of the primer sets varied from 98.5% (*GAPDH*) to 106.9% (*EF1A*), and correlation coefficients ranged between 0.987 and 0.999 (Table 1). All efficiencies were considered acceptable for use. We strongly recommend the calculation of primer efficiencies for use qPCR-based differential gene expression analyses.

### Expression of reference genes across samples

To determine reference gene expression across a wide set of conditions, the steady-state levels of all nine genes were analyzed in cDNA samples from fifteen different treatments (in biological triplicate) with two time-points each (3 hours and 3 days post-treatment): auxin, gibberellic acid, EtOH control, imidacloprid, $CuSO_4$, hypersaline 2x artificial seawater (ASW), hyposaline

**Table 1. Description statistics of housekeeping gene candidates.**

| Gene name | Gene symbol | primer F | primer R | Fragment length | Tm [°C] | PE[%] | r2 |
|---|---|---|---|---|---|---|---|
| *Elongation factor 1 alpha* | *EF1A* | ATGAGGTGGCCATCTACCTG | CCCTTGTACCAAGGCATGTT | 124 | 59.85 | 106.90 | 0.999 |
| *14-3-3* | *14-3-3* | CGAGACAGAGTTGACGGACA | CGCAAGATACCGGTGGTAGT | 133 | 60.02 | 99.76 | 0.999 |
| *Actin* | *ACT* | GACCTTTACGGCAACATCGT | GGTGCCACAACCTTGATCTT | 122 | 59.97 | 106.86 | 0.989 |
| *Ubiquitin conjugating enzyme 2* | *UBCE2* | AAGCTCAACATGGGCTGTGT | GCCACCAGTACCTGCTCAAT | 110 | 60.14 | 103.52 | 0.995 |
| *Ribosomal subunit 40S* | *40s* | ACGGCTGTCTGAACTTCACC | ACCTTCACCACCTTGAAACG | 112 | 60.01 | 105.70 | 0.989 |
| *Elongation factor 1 beta* | *EF1B* | TTCGGAGTGAAGAAGCTCGT | CAGAGGCGGTTCATCGTAGT | 134 | 60.28 | 104.70 | 0.997 |
| *Elongation factor 2 alpha* | *EF2A* | TGGACCACGGAAAGTCTACC | GGTGATACATCGGTCCTGCT | 125 | 59.96 | 106.34 | 0.996 |
| *Actin related protein complex* | *ARP2/3* | GGAAGCCTCTGGCTATTGGT | GTGGTCTTGGCTTGGAACAT | 123 | 59.97 | 105.05 | 0.998 |
| *Glyceraldehyde-3-phosphate dehydrogenase* | *GAPDH* | TCTTGGGTTACACCGAGGAC | GTACCACGACACGAGCTTGA | 125 | 59.9 | 98.43 | 0.987 |

$T_m$, melting temperature; P*E*, PCR efficiency; r$^2$, correlation coefficient.

0.5X ASW, desiccation, wounding, high light, low light, high temperature, low temperature, ASW + Provasoli nutrients, and ASW with no additional nutrients as the control (detailed in Materials and Methods; S1 Table). The quantification cycle (Cq) was identified for each gene in the triplicate samples by qPCR. The Cq values of all tested genes lay between 18.63 and 28.21 (except a single replicate outlier in *EF2A* at Cq = 32.7) (S4 Fig). The most highly expressed gene was *EF1A* (Cq$_{mean}$ = 20.6), followed by *40S*. The lowest expression level was observed for *GAPDH* (Cq$_{mean}$ = 26.9) (Fig 1). For each gene, the variation across samples did not exceed 3 cycles, which suggests a relatively constant expression in all conditions. All nine candidate genes were determined as suitably 'consistent' (qualitatively) across conditions to move forward in our analyses.

## Reference gene expression stability in all samples

The stability of expression for a given reference gene is a quantitative measure by which the suitability of the reference gene may be assessed. Stability refers to how invariant the expression pattern of a given gene is in an experimental setup. The stability of our nine candidate reference genes was evaluated across all 15 conditions and 2 time-points (Fig 2). Stability was analyzed using three algorithms: geNorm [8], NormFinder [11], and BestKeeper [12]. These algorithms rank the reference genes according to the calculated gene expression stability values and acceptability threshold (geNorm (M<1.5); NormFinder (SV<1.0)) or standard deviation and threshold (SD<1.0, BestKeeper) (detailed in Materials and Methods). Fig 2 shows the calculated stability metric for each gene in all samples combined, by method, ranked from best to worst (left to right).

The rankings generated by geNorm and NormFinder were similar; *EF1B* and *GAPDH* were identified as the most stable genes, followed by *ACT*, *14-3-3*, *EF1A*, and *40S* (Fig 2A and 2B). BestKeeper, on the other hand, ranked *40S* and *14-3-3* as the most stable (Fig 2C). The three least stable genes identified by all three algorithms were *UBCE2*, *EF2A*, and *ARP2/3* (Fig 2A–2C). To create a consensus of the best-to-worst gene ranking across all three programs, a rank aggregation method was performed using a Cross Entropy Monte Carlo algorithm (R-package RankAggreg; See Materials and Methods). This analysis indicated that *GAPDH* and *EF1B* were the two most stable genes and *EF2A* and *ARP2/3* the most unstable (Fig 2D). Based on these analyses, *GAPDH* and *EF1B* are recommended as general normalization genes for qPCR gene expression studied in *Fucus*.

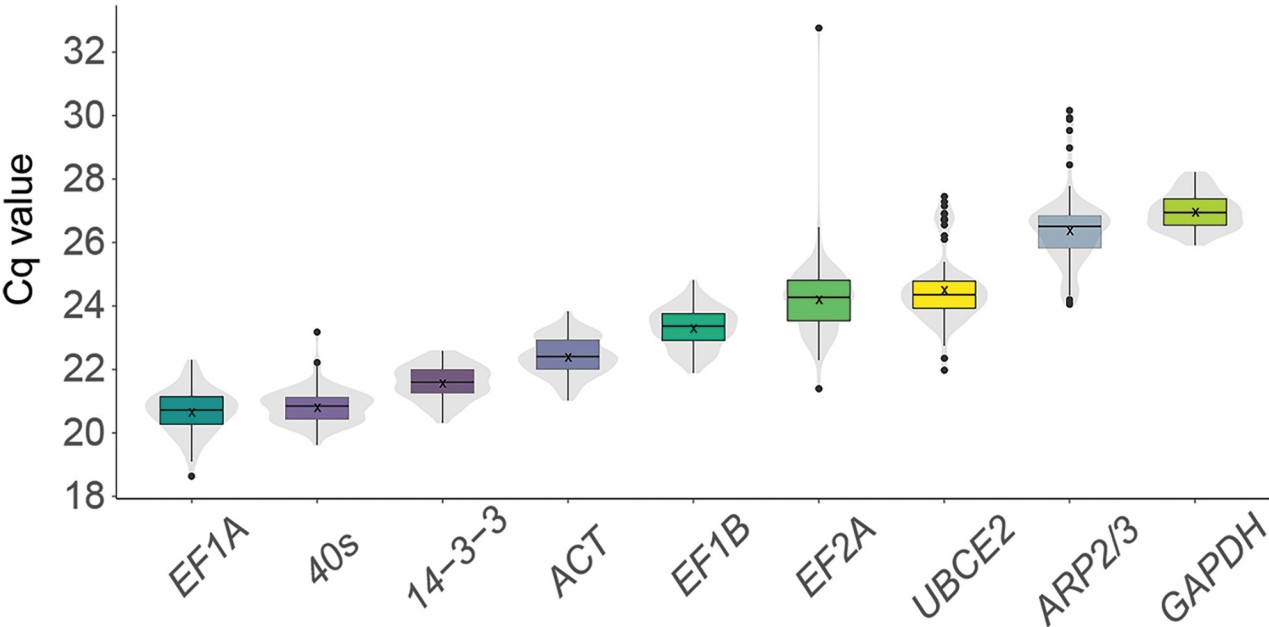

**Fig 1. Expression level of tested housekeeping genes.** The distribution of Cq (quantification cycle) values of 9 reference genes pooled across 30 samples (15 experimental conditions x 2 time-points) obtained using qPCR. The boxplot marks the median (line) and 25th (lower) and 75th (upper) percentile; x marks the mean; the underlying violin plots show the data distribution for each housekeeping gene. Outliers are depicted as black dots. Data for individual experimental conditions may be found in S4 Fig.

### Reference gene expression stability in sample groups (conditions)

Numerous reports acknowledge the importance of using an appropriate reference gene for a given experiment since individual reference genes may not maintain normalized expression in all conditions [10,47,48]. To test which reference genes would be most suitable for each of our stress-condition groups, expression stability was analyzed per group using geNorm, NormFinder, and BestKeeper. The samples were grouped based on the nature of their stressor as follows: *Fucus* is an intertidal alga that is challenged by a very harsh natural environment where it undergoes daily desiccation and osmotic shock (group 1 –physiological stress); furthermore, these environments are under the constant pressure of pollutants, such as copper and herbicides/pesticides (group 2—pollution); also, brown algae serve as food for marine organisms such as mollusks, so the effect of grazing (proxied by mechanical wounding) was examined as well (group 3 –wounding); in Stramenopiles, phytohormones have been found to influence developmental processes [49–51] and as such we have tested the effect of auxin and gibberellic acid (group 4—hormones); *Fucus* was also grown with and without nutrients to identify the best housekeeping genes for nutritional studies (group 5—nutrients); lastly, we cultured *Fucus* under different light and temperature regimes (group 6 –temperature-light). The ranking of reference genes in each treatment group, based on stability, is shown in Fig 3. The Cq values for each reference gene, by treatment group, can be found in S4 Fig. The stability values for each gene by treatment group, per algorithm, can be found in S5 Fig.

GAPDH was identified as one of the top three most stable genes across multiple conditions (4 out of 6) and ARP2/3 as one of three least stable genes (4 out of 6), whereas the stability of other candidate reference genes varied depending on the condition examined (Fig 3; S5 Fig). As mentioned earlier, the three algorithms rank reference genes according to their calculated gene expression stability values and acceptability thresholds (geNorm: M<1.5; NormFinder: SV<1.0) or standard deviation (BestKeeper: SD<1.0). Based on these thresholds, in the

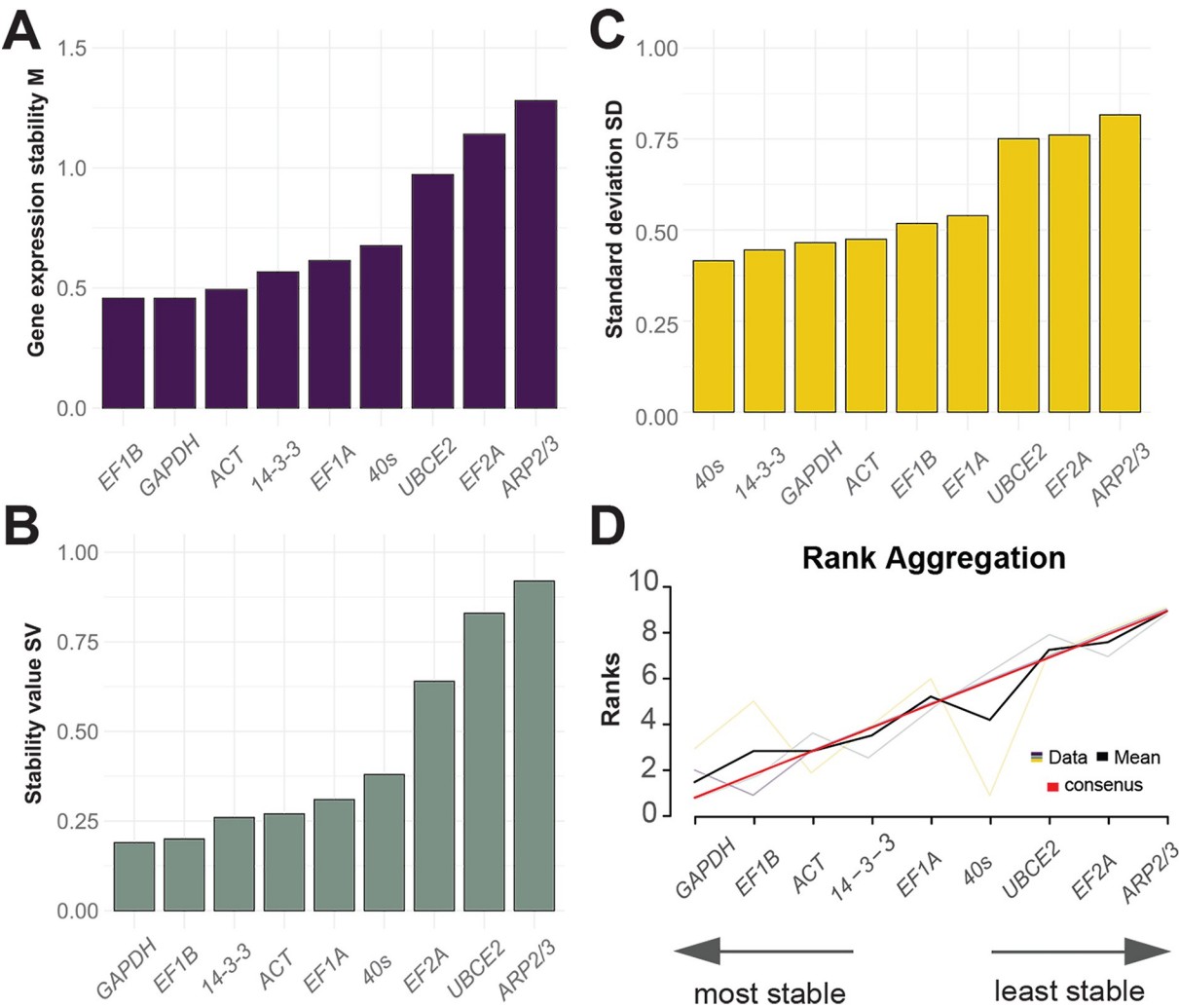

**Fig 2. Individual ranking of housekeeping genes by stability.** Stability values for nine candidate reference genes generated by the following algorithms: A) geNorm, B) NormFinder, C) BestKeeper, and D) a consensus (by rank aggregation). The maxiumum scale for A-C represent the maximum acceptability value for each stability metric.

hormone and the temperature-light groups, *ARP2/3* and *UBCE2* were rejected as suitable reference genes (S4 and S5 Figs).

geNorm, BestKeeper, and NormFinder showed differences in the ranking of candidate reference genes within treatment groups, to some extent (Fig 3). To achieve a consensus ranking across all three methods that could be used for recommendation of normalization genes by condition, we again performed a rank aggregation analysis (Fig 3, red line and left to right order). For pollution, hormone, nutrient and temperature-light treatments there was less variation in ranking by the three algorithms and the consensus appears to be a reasonable recommendation. However, for physiological stress and wounding, the results of the three algorithms differed more, and the consensus seemed qualitatively less reliable. As such, for these two conditions, we recommend using the ranking produced by geNorm or NormFinder; our recommendation here is based on geNorm (See Discussion).

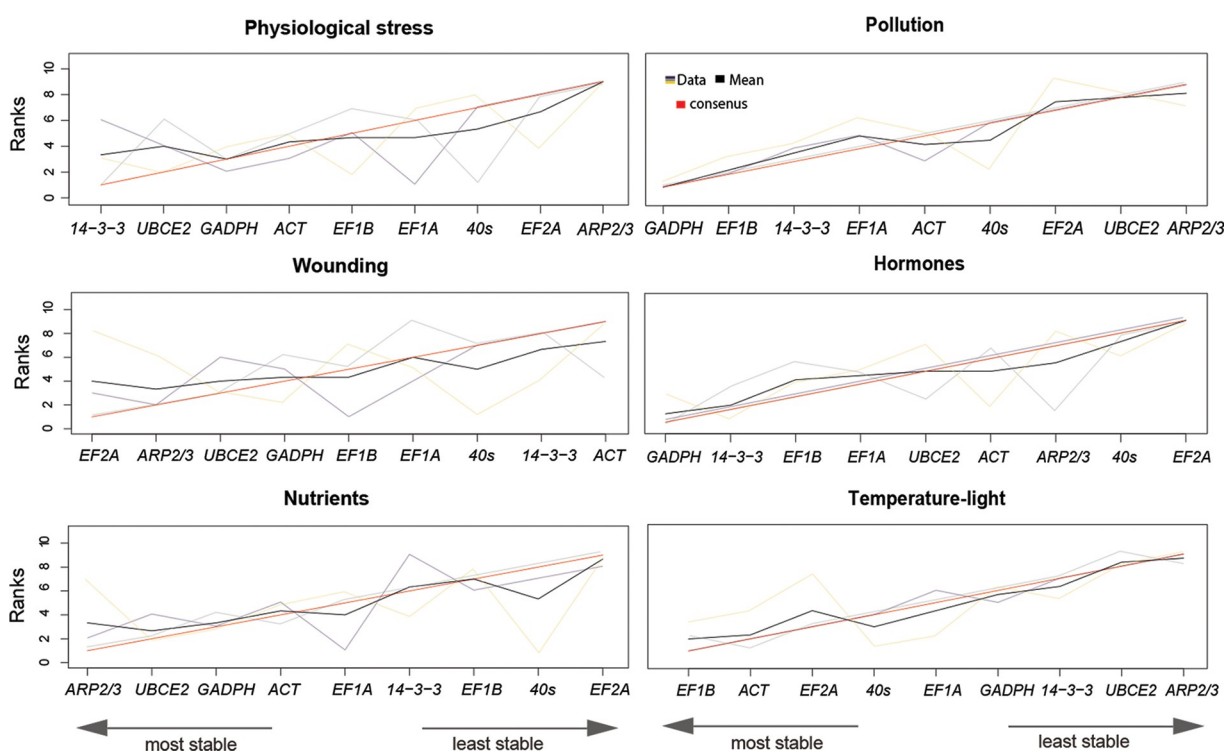

**Fig 3. Optimal stability ranking of candidate reference genes using geNorm, NormFinder, BestKeeper, and rank aggregation method.**
Gene ranks from the three stability algorithms are shown by treatment group. Line colors represent: purple (geNorm), green (NormFinder),
yellow (BestKeeper), black (mean rank), and red (consensus rank).

## Identifying the optimal number of reference genes for normalization

Our previous analyses allowed us to provide recommendations for normalization genes to be
used in general and specific experimental designs. However, using a single reference gene dur-
ing qPCR experiments can cause bias and the use of multiple reference genes is suggested as
standard practice [8]. To determine the optimal number of reference genes to be used in *Fucus*
qPCR experiments, pairwise variation $V_{n/n+1}$ was calculated for the proposed reference genes
using the geNorm algorithm [8]. In this method, the reference gene variances were combined
additively and successively according to geNorm rank. After each addition the pairwise vari-
ance was calculated between that sum ($V_{n+1}$) and the variance of the prior sum ($V_n$); for exam-
ple, $V_{2/3}$ was the pairwise variance calculated when using the top two genes was compared
with using the top three. In general, the more reference genes that are added, the lower the suc-
cessive pairwise variance becomes. When the pairwise variance was below 0.15, *n* number of
normalization genes was considered sufficient.

When all conditions were pooled, to simulate a general experiment, $V_{2/3}$ was below the 0.15
threshold, suggesting that using the top two stable reference genes was sufficient (Fig 4; $V_{2/3}$ =
0.145; optimal reference gene set *GAPDH+EF1B*). When the analysis was performed by condi-
tion group, $V_{2/3}$ was below 0.15 for all conditions except pollution (physiological stress = 0.130,
hormones = 0.096, nutrients = 0.103, temperature-light = 0.120, wounding = 0.105, pollu-
tion = 0.173; Fig 4). For pollution stress, $V_{3/4}$ was less than the threshold ($V_{3/4}$ = 0.145) suggest-
ing the need for three reference genes for these experiments (Fig 4). Taken together, two stable
reference genes are sufficient for most conditions tested here. By combining the recommended
number of reference genes from this analysis with the consensus rank achieved for stability, we
generated a table of most recommended and least recommended reference genes (Table 2).

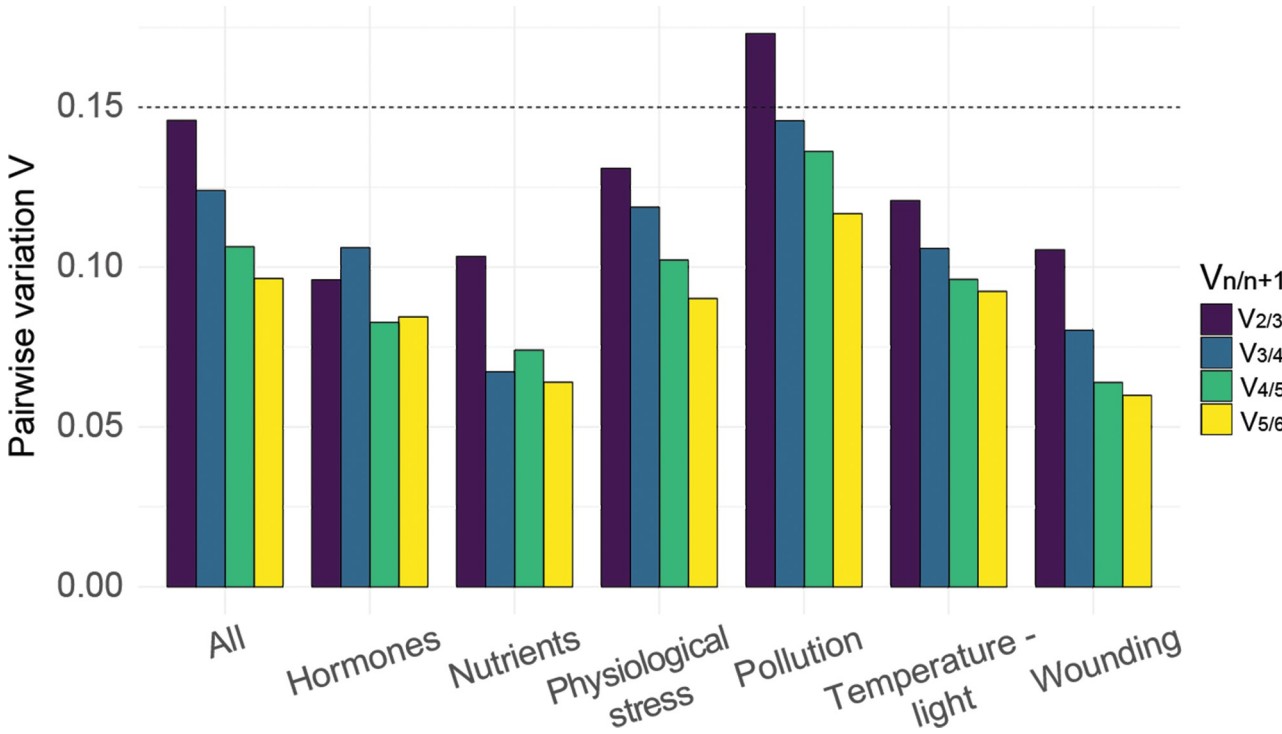

**Fig 4. Calculation of optimal number of housekeeping genes.** Pairwise variation ($V_{n/n+1}$) was calculated in all tested samples; all (all conditions together), hor (hormones), nutr (nutrients), phy (physiological stress), pol (pollution), t-l (temperature-light), wou (wounding). The $V_{n/n+1}$ values below the 0.15 threshold (dotted line) indicate that $n$ normalization genes are sufficient.

**Table 2. List of recommended reference genes for each of the tested conditions.**

|  | Reference gene pair with highest stability | Reference gene pair with lowest stability |
|---|---|---|
| All samples | *GAPDH* | *EF2A* |
|  | *EF1B* | *ARP2/3* |
| Physiological stress | *GAPDH* | *EF2A* |
|  | *EF1A* | *ARP2/3* |
| Pollution | *GAPDH* | *UBCE2* |
|  | *EF1B* | *ARP2/3* |
|  | *14-3-3* |  |
| Wounding | *EF1B* | *14-3-3* |
|  | *ARP2/3* | *ACT* |
| Hormones | *GAPDH* | *40S* |
|  | *14-3-3* | *EF2A* |
| Nutrients | *ARP2/3* | *40S* |
|  | *UBCE2* | *EF2A* |
| Temperature—light | *EF1B* | *UBCE2* |
|  | *ACT* | *ARP2/3* |

The recommendation is based on the result of a rank aggregation consensus gene stability analysis (pollution, wounding, hormones, nutrients, temperature-light) and geNorm stability analysis (physiological stress).

## Validation of reference genes in conditions of physiological stress

To experimentally validate our reference gene recommendations, we performed qPCR for two heat shock proteins, *Hsp70* and *Hsp90*, in samples from *Fucus distichus* under physiological stress (salinity and desiccation). *Hsp70* and *Hsp90* were chosen as potential target genes as they had exhibited differential expression in *Fucus* under various stress conditions [40,42,52]. *Hsp70* and *Hsp90* sequences were identified in the *F. serratus* embryo transcriptome [36] and primers were designed and tested (S6 Fig). We then examined gene expression using the ΔΔCq method with 1) normalization to the two most stable reference genes (*GAPDH* and *EF1A*) or 2) with normalization to the least stable genes (*EF2A* and *ARP2/3*) for physiological stress (Table 2). Gene expression analysis with the most stable pair (*EF1A* + *GAPDH)* showed a significant increase of *Hsp70* cDNA levels with 0.5x ASW treatment and a significant decrease in levels with the 2x ASW stress and desiccation treatments (Fig 5A). Conversely, normalizing to the *ARP2/3/EF2A* (least stable) reference gene set resulted in a loss of significant differential expression in the 0.5X ASW treatment (Fig 5B). *Hsp90* cDNA levels did not show significant change with treatment using either normalization set (Fig 5). These data indicate that using the most stable pair of normalization genes allowed for the capture of more information on differential gene expression for *Hsp70*; however, even the lowest-ranked normalization pair did result in some differential expression detection. Using a suboptimal normalization pair would likely be most detrimental when high Cq variance within technical replicates or between biological replicates is present. As such, using the most recommended set of normalization genes, in general or by condition, is likely to increase experimental sensitivity and reduce false negatives.

## Discussion

The use of stable and suitable reference genes when analyzing qPCR data is of utmost importance to correctly assess, present, and interpret gene expression in any experimental design. In this study, we analyzed the suitability of nine candidate genes for normalization of gene expression data obtained from qPCR in a brown alga *Fucus distichus*. This paper reports recommendations for reference gene pairs to be used in *Fucus* experimental qPCR studies (Table 2). The recommendation is a result of two analyses: stability assessment and calculation of an optimal number of reference genes to be used. Below we discuss the outcomes and comparative merits of these analyses.

### Assessing stability

The expression stability of the candidate genes was tested using the geNorm, NormFinder, and BestKeeper algorithms. Our analysis with the three algorithms showed that the stability of most candidate genes strongly depended on the experimental condition used, with *GAPDH* being the most stable gene in most conditions and *ARP2/3* being the least stable in most conditions. We did observe some discrepancies in ranking candidate gene stability within conditions when using these three different algorithms.

The differences in stability values observed between different algorithms were not entirely unexpected, taking into account that there are differences inherent in the three approaches [8,11,12]. geNorm is based on the principle that the most stable housekeeping genes should have a nearly identical variation in expression ratio, or co-expression pattern. This approach, however, may identify co-differentially expressed genes as highly stable in the chosen experimental system [8,11]. Conversely, NormFinder is not affected by the co-differential expression problem and as such should be more robust; it estimates both inter- and intra-group variation and then combines them into a stability value [11]. BestKeeper, on the other hand, uses raw

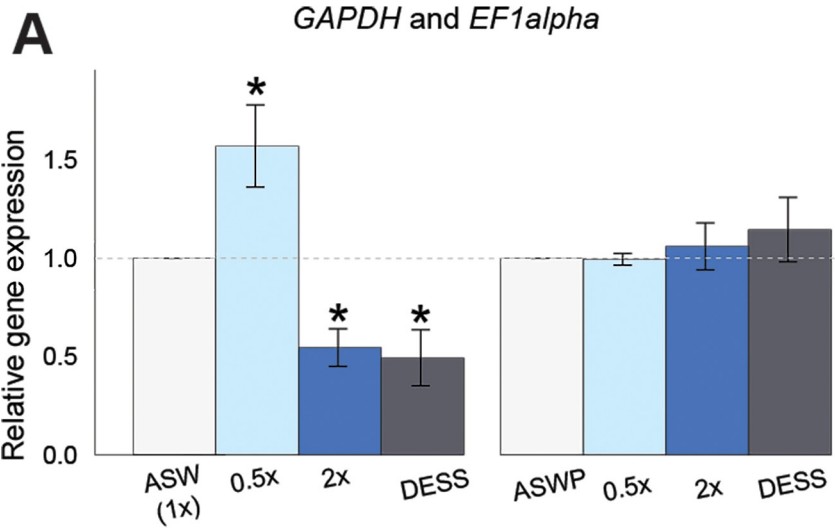

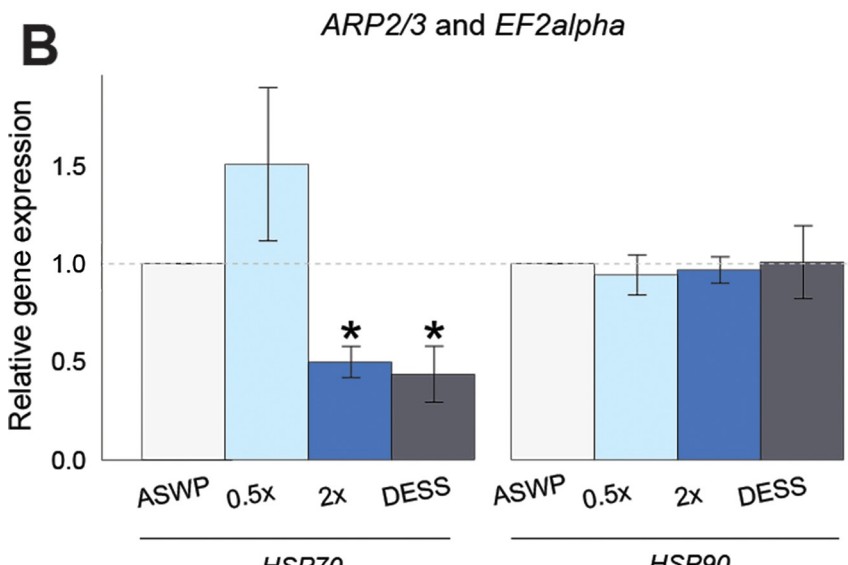

**Fig 5. Detection of significant change in gene expression depending on the reference gene choice.** Expression of two heat shock protein genes (*Hsp70* and *Hsp90*) normalized by the two most stable (A: *EF1A* and *GAPDH*) and two least stable (B: *ARP2/3* and *EF2A*) reference gene pairs. Statistically different gene expression from ASWP is indicated by * (Student's t-test; normal distribution, unequal variance; $p < 0.05$).

Cp values as input to identify the best among the investigated candidate genes. It uses a Pearson correlation analysis, a parametric method, which is valid for normally distributed data with a homogeneous variance; if the data do not match these dependencies it may lead to a false interpretation of the obtained results.

When the stability of genes in all the treatments was tested together, some discrepancies were detected in the ranking of the most stable candidate reference genes using the three algorithms, but there was an agreement between ranking the least stable genes. *GAPDH* and *EF1B* were ranked as the most stable housekeeping genes tested by geNorm and NormFinder, however, BestKeeper only placed them 3rd and 5th, respectively. The least stable genes were identified as *ARP2/3*, *EF2alpha*, and *UBCE*. This is contradictory to results in *Ectocarpus siliculosus*,

where both *ARP2/3* and *UBCE* were the most stable genes and *GAPDH* was one of the most unstable genes [13]. It may be that patterns of gene expression vary between different brown algal species, which further emphasizes the need to precisely define best reference genes for specific studies.

According to our results, in four out of seven tested conditions *GAPDH* and *EF1B* were the most stable reference gene candidates. It is valuable to note that *GAPDH* and *EF1B* had different Cq$_{mean}$ values (26.9 and 20.6, respectively) providing candidate normalization genes at different expression level tiers. These genes have been previously identified as stably expressed in other systems such as green algae, brown algae, plants, and animals [14,15,17,18,53,54]. Some studies in *Fucus* directly selected common reference genes such as *EFAs* to normalize their target genes [41,42,45]. *EFA* genes have been one of the most widely used in normalizing gene expression of algal and plant species under stress conditions [13,17,22,54,55]. However, *EFAs* are not necessarily optimal for all stress conditions in *Fucus*, as shown here.

Some of the candidate genes were mostly present in the middle of the stability range, such as *14-3-3*, *40S*, and *actin*. *Actin* and ribosomal subunit or RNA genes have previously been reported as variably expressed in algae [13,16] and other organisms [17,22,56,57]. The most unstable genes from our study were *ARP2/3*, *UBCE2*, and *EF2A*. Even though their stability ranking position varied depending on the treatment, for all of the conditions except wounding, at least one of the aforementioned genes was ranked last. *ARP2/3* or *UBCE* have previously been used as normalization genes in the brown alga *Ectocarpus siliculosus* and were stably expressed in that system [13], consistent with our analysis. In line with our results, some previous reports in algae and plants identified *UBCE* as a variable and non-suitable reference gene [16,54].

What is perhaps most pertinent to note is that even though ranking differences were observed between the different algorithms, the majority of the calculated stability values were still suitable as the fell under the rejection threshold (geNorm >1.5, BestKeeper>1, NormFinder>1; S5 Fig). This suggests that, generally, all nine of the proposed reference genes we started with should be suitable for normalization. We note that there are specific stress conditions where a handful of candidate genes were rejected for use (hormone and temperature-light; S5 Fig) and so care in specific stress conditions should be taken when selecting the exact pair of normalization genes.

## The optimal number of reference genes

It has been shown that the inclusion of more than one reference gene is required to detect subtle changes in gene transcript levels by qPCR analysis [8,11,58]. Vandesompele et al. [8] provide evidence that normalization using a single gene can lead to an inaccurate normalization of up to 25% of cases (with up to 6.4-fold difference). In our experiments and analyses, we could conclude that in all treatment groups, except pollution, using the two most stable genes should be sufficient for normalization. For the pollution treatment, our experiment would require three normalization genes according to analysis. We strongly recommend that experimental designs for *Fucus* RT-PCR and qPCR include at least two reference normalization genes.

## Evaluating normalization pairs in a desiccation and salinity stress study

Overall, our recommendation of a 'best' normalization pair for physiological stress condition (*GAPDH* and *EF1A*) proved to be more sound when compared to the 'least' suitable pair when evaluating the differential gene expression of *Hsp70* during salinity stress and desiccation. The statistically significant difference in the expression of *Hsp70* in low salinity failed to be detected

when normalizing to the two least stable genes. This has previously been shown in other reports as well, where the choice of unstable genes led to misinterpretation of expression data [47,59,60]. It should be noted that we have assumed that the differential expression of *Hsp70* in low salinity is true and not a false positive; we believe this to be the most likely case given the higher stability of the 'best' pair.

## Conclusion

The selection of suitable reference genes to test qPCR based gene expression is a necessary first step in understanding key molecular networks in the brown algal lineage—here we recommend the usage of specific reference gene sets in specific conditions when performing comparable experiments in the brown alga *Fucus distichus*. In general, at least two reference genes are recommended, and the best 'general' pair to use are *GAPDH* and *EF1B*. When designing new experiments, we recommend checking the top five general recommended genes from this study (Table 2) to test their stability in one's experimental system, as conducted here, before deciding on the best normalization gene set for your study.

## Materials and methods

### Culture conditions and experimental design

*Fucus distichus* (Fucales, Phaeophyceae) individuals were collected from their natural rockpool habitat at the University of California Kenneth S. Norris Rancho Marino Reserve (Cambria, CA). Multiple apical segments were cut from adult individuals and cultivated in 2L glass flasks in a Percival incubator (Percival Scientific, USA) at 16˚C in filter-sterilized artificial seawater (ASW; 450 mM NaCl, 10 mM KCl, 9 mM CaCl$_2$, 30 mM MgCL$_2$·6H$_2$O, 16 mM MgSO$_4$·7H$_2$O) enriched with Provasoli medium (PES) for acclimation [61]. Samples were cultured under a white fluorescent light at 60 μmol m$^{-2}$ s$^{-1}$ with a 12h:12h light: dark cycle. After 7 days, individual apical segments were transferred into Petri dishes with 15 different treatments, one segment per condition (S1 Table). Three biological replicates were obtained for each treatment at two time-points: 3 hours and 3 days after treatment start, resulting in 90 samples in total. A summary of the treatments may be found in S1 Table.

The chemical treatments were: control ASW with nutrients (PES), nutrient-deficient ASW, 0.2 μg/L imidacloprid (Marathon 1%, OHP, Inc., USA), 50 μM indole-3-acetic acid (IAA; Cat#102037, MP Biochemicals, Irvine, CA.), 50 μM gibberellic acid (GA; Cat#G7645, Sigma). An equal volume of absolute ethanol was used as a control for the GA and IAA treatment. In addition, a saline shock was performed using a hypersaline solution (2x ASW) and a hyposaline solution (0.5x ASW). The desiccation treatment was affected by placing the algal segments onto dry Petri dishes, after blotting gently, under the same environmental conditions as in other treatments. A mechanical wounding treatment, to simulate the effect of grazing, was performed by damaging the algal segments with a razor blade in several places along the thallus. Alteration of light and temperature was achieved as follows: a portion of samples was cultured under modified light conditions (120 μmol m$^{-2}$ s$^{-1}$ and complete darkness) or two non-standard temperatures (8˚C and 22˚C).

### RNA extraction and cDNA synthesis

Tissue was flash frozen in liquid nitrogen after treatment (3 hours and 3 days) and immediately ground in liquid nitrogen with a pestle and a mortar. In our hands, extraction of RNA from *Fucus* tissue is impaired by storage of the intact and/or ground tissue at -80 for longer periods of time. To further refine the tissue homogenate samples were ground in 3 mL Duall

glass grinders (Cat# K885451/0021, Smith Scientific, UK) grinder in 1ml of CTAB buffer (2% CTAB, 100 mM Tris-HCl, 1.5 M NaCl, 50 mM EDTA, 50 mM DTT). RNA extraction was adapted from Apt et al. [62] as follows briefly. Samples were shaken on a tilt shaker at room temperature for 20 minutes after which 1v of chloroform was added to each. Solutions were mixed by inverting the tubes several times and additionally incubating for 20 minutes while gently shaking. Samples were then centrifuged at 10,000g for 20 minutes at 4˚C, after which 0.3v of 100% ethanol was added to remove the polysaccharides. Polysaccharides were further extracted with 1v of chloroform and centrifuged for 20 minutes at 10,000g at 4˚C. The upper phase was transferred to a new tube and RNA was precipitated overnight at -20˚C by adding 0.25v LiCl and 1% (v/v) beta-mercaptoethanol. Samples were centrifuged for 30 minutes 13,000g (4˚C), after which the pellet was re-suspended in 50 µl of DEPC-treated MiliQ water. To remove residual DNA, a DNase treatment was performed (TURBO DNase, ThermoFisher) according to manufacturer's instructions. The final volume was adjusted to 500 µl with RNAse free water and extraction was performed by adding 1v of phenol: chloroform: isoamylic alcohol (25:24:1 v/v). The samples were centrifuged at 10,000g for 20 minutes (4˚C), after which another 1V chloroform extraction was carried out and centrifuged again. The upper phase was transferred to a clean RNAse free tube and RNA was precipitated by addition of 0.1v sodium acetate (pH5.5) and 2.5v of 100% ethanol overnight at -20˚C. After centrifugation (30 minutes at full speed, 4˚C), the supernatant was carefully aspirated with a pipette and the pellet was washed with 75% ethanol. The tubes were centrifuged for 10 minutes at full speed (4˚C). The supernatant was carefully removed and the tubes were left to air dry for 5–10 minutes. The pellet was then re-suspended in RNAse free water.

The purity of RNA was assessed by measuring the ratio $OD_{260}/OD_{280}$ and $OD_{230}/OD_{260}$ using a NanoDrop 2000 (Thermo Fisher Scientific, USA). RNA integrity was measured using the High Sensitvity RNA ScreenTape Assay in an Agilent TapeStation (Agilent Technologies, Inc.; S7 Fig). The RNA sample (40 ng) was reverse-transcribed to cDNA using RevertAid First Strand cDNA Synthesis Kit with oligo (dT)20 primers (Thermo Fisher Scientific, USA) according to the manufacturer's instructions. cDNA samples were diluted to 10 ng/µl for qPCR and kept at -20˚C until further use.

## Quantitative real-time PCR (qPCR)

For each candidate gene, a pair of oligonucleotide primers was designed using Primer3 Input (v. 4.1.0) online tool (Table 1). qPCR was performed using a CFX384 Real Time PCR System (Bio-rad Laboratories Inc., USA). For each test, 3 µl of cDNA (in technical duplicate wells) template was amplified using the SsoAdvanced™ SYBR® Green Supermix (Bio-rad Laboratories Inc., USA) in a final volume of 15 µl to test housekeeping gene expression levels. The cycling was performed as follows: 95˚C for 5 min followed by 41 cycles of 30s at 95˚C, 30s at 60˚C and 30s at 72˚C and a final step of 95˚C for 1 min. Each run was finished with heating up the samples from 65˚C to 95˚C to obtain a melting curve to test the specificity of amplification. All of the amplicons tested here had single melting peaks indicating a unique amplification product (S2 Fig).

Primer efficiencies were calculated as follows: a pooled cDNA sample of all samples was mixed and a dilution series generated to yield 1x, 0.1x, 0.01x, and 0.001x dilutions. Each primer set was used to amplify from the dilution series using the conditions above. The primer amplification efficiency (PE) and the correlation coefficient (R2) of each primer pair were calculated (Table 1, S3 Fig). All of the primer sets reported here fell within accepted boundaries of PE and R2. Cq for the NTC for each primer set were all above 35 cycles (*EF1A* = 37.71, *EF2A* = 39.44, *EF1B* = undetected, *GAPDH* = 38.33, *UBCE2* = 39.51, *ACT* = 35.78, *14-3-3* =

undetected, *40s* = 38.39, *ARP2/3* = undetected). A MIQE checklist (https://rdml.org/miqe.html) is provided in the supplement (S2 Table).

## Assessing the stability of candidate gene expression

Data stability analysis was performed on all samples (15 conditions x 2-time points x 3 biological replicates) for all of the 9 genes identified, as well as separate groups with respect to the nature of the treatment (physiological stress, hormone addition, pollution stress, temperature-light modification, nutrient modification, and wounding/grazing (S1 Table). Quantification cycle (Cq) values were analyzed with geNorm, NormFinder, and BestKeeper.

GeNorm calculates an average expression stability value (M) for each reference gene. The analysis allows ranking of the genes according to their expression stability based on an iterative stepwise exclusion of the genes with the highest M value (lowest stability). Additionally, it calculates the optimal number of reference genes to be used for normalization, through a pairwise variation (V) test [8]. NormFinder calculates the expression stability value for each gene using ANOVA-based mathematical analysis, taking into account intra- and inter-group variations of the samples. A low SV-value indicates the high expression stability of this gene [11]. BestKeeper is an Excel-based tool that uses the standard deviation (SD) and the coefficient of variance (CV) as evaluation criteria for stably expressed reference genes. Stability values for geNorm were analyzed using R package NormqPCR [63] (NormFinder stability values were calculated using the NormFinder R script, and BestKeeper Excel-based analysis was performed and standard deviation (SD) was used as a stability measure [12]. In addition, a rank aggregation method based on a Monte Carlo cross-entropy algorithm (R-package; RankAggreg 0.6.5; https://CRAN.R-project.org/package=RankAggreg; with default settings and following set parameters: method = "CE", distance ="Spearman", convIn = 7, seed = 100) was used to combine the gene ranks of the three above algorithms and create a consensus housekeeping gene ranking.

## Validation of chosen housekeeping genes

Apical segments were placed the following physiological stress conditions: 2x salinity, 0.5x salinity, and desiccation condition for 6 hours. Untreated samples were transferred to 1X ASW. After 6 hours, RNA was extracted as detailed above and gene expression analyzed. To test the normalization efficiency of reference genes in a specific condition (ΔΔCq method), two of the most stable and two of the most unstable candidates for this group (physiological stress) were used to test the expression level of two heat-shock proteins (*Hsp*). Hsp gene sequences were identified through a local BLAST of the *Ectocarpus Hsp70* and *Hsp90* to the *Fucus serratus* transcriptome [36] and primers were designed using Primer3 Input (v. 4.1.0) online tool. ΔΔCq was calculated by normalization of the *Hsp* genes with the two housekeeping genes and to the targeted gene expression detected in a separate control sample.

## Supporting information

**S1 Fig. Sequences of all used reference genes.** Each reference gene was matched to a *Fucus* transcriptome gene (Trinity gene CDS; Linardić et al., 2020) and PCR amplicons resulting from the PCR amplification with the designed primers in Table 1.
(PDF)

**S2 Fig. RT-qPCR Melt curves of 9 candidate housekeeping genes.** -ΔF/ΔT (change in fluorescence/change in temperature) is plotted against temperature to obtain a clear view of the

melting dynamics of each reference gene.
(PDF)

**S3 Fig. PCR amplification efficiency curves.** Amplification efficiency is determined from the slope of the log-linear portion of the calibration curve (y function coefficient). The initial template concentration (the independent variable; $\log_{10}$ of dilution 1x, 0.1x, 0.01x, 0.001x) is plotted on the x axis and corresponding Cq (the dependent variable) is plotted on the y axis.
(PDF)

**S4 Fig. Reference gene expression in individual condition groups for all 9 reference genes, with consistent coloring by gene name.** The boxplot marks the median (line) and 25th (lower) and 75th (upper) percentile; x marks the mean; the underlying violin plots show the data distribution for each housekeeping gene. Outliers are plotted as black dots.
(PDF)

**S5 Fig. Stability analysis values of candidate reference genes in the 6 stress condition groups.** Stability values are are conditionally formatted (shade of color) according to their stability value, from the most stable (darkest shade) to the least stable (lightest shade). Purple corresponds to geNorm, green to NormFinder and yellow to BestKeeper analysis. Red text indicates genes over the threshold of acceptability, by algorithm.
(PDF)

**S6 Fig. Primers and sequences for *Hsp70* and *Hsp90* used in the study.**
(PDF)

**S7 Fig. TapeStation analysis of the 90 RNA samples.** The High Sensitivity RNA ScreenTape assay was used for analyzing and assessing integrity of total RNA in all samples (following the manufacturer's instructions) on a 2200 TapeStation Bioanalyzer (Agilent Technologies Inc.). RIN = RNA integrity number.
(PDF)

**S8 Fig.**
(JPG)

**S1 Table. Table of conditions used for housekeeping gene analysis.** Column 1 –treatment, column 2 –parameters of the treatment, column 3 –length of treatment, column 4 –associated group of treatments.
(PDF)

**S2 Table. MIQE checklist (https://rdml.org/miqe.html) for the qPCR experiment.**
(PDF)

## Acknowledgments

We thank the developers of geNorm, NormFinder, and BestKeeper for their tools. We also thank Dr. Lauren Dedow for critical reading of the manuscript. We thank the members of the Braybrook and Pellegrini labs for their comments and support during this project.

## Author Contributions

**Conceptualization:** Marina Linardić.

**Data curation:** Marina Linardić.

**Formal analysis:** Marina Linardić.

**Funding acquisition:** Siobhan A. Braybrook.

**Investigation:** Marina Linardić.

**Methodology:** Marina Linardić.

**Project administration:** Marina Linardić, Siobhan A. Braybrook.

**Supervision:** Siobhan A. Braybrook.

**Validation:** Marina Linardić.

**Visualization:** Marina Linardić.

**Writing – original draft:** Marina Linardić.

**Writing – review & editing:** Marina Linardić, Siobhan A. Braybrook.

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
