## [Decision Letter · Decision Letter 0]

17 Aug 2020

PONE-D-20-12656

Identification and selection of optimal reference genes for qPCR-based gene expression analysis in Fucus distichus under various abiotic stresses

PLOS ONE

Dear Dr. Siobhan A. Braybrook,

Thank you for submitting your manuscript to PLOS ONE. After careful consideration, we feel that it has merit but does not fully meet PLOS ONE’s publication criteria as it currently stands. Therefore, we invite you to submit a revised version of the manuscript that addresses the points raised during the review process.

Aside from the format/typo issues indicated by referee the authors must state the parmamters applied for RankAggreg (CE Kendall or CE Spearman? weight settings? what stopping criteria?) to ensure replicability. A MIQE checklist (https://rdml.org/miqe.html) should be added as supplement information and experimental details supplied accordingly.

We look forward to receiving your revised manuscript.

Kind regards,

Christian Schönbach, Dr.rer.nat.

Academic Editor

PLOS ONE

Journal Requirements:

Reviewers' comments:

Reviewer's Responses to Questions

**Comments to the Author**

1. Is the manuscript technically sound, and do the data support the conclusions?

Reviewer #1: Yes

2. Has the statistical analysis been performed appropriately and rigorously? 

Reviewer #1: Yes

3. Have the authors made all data underlying the findings in their manuscript fully available?

Reviewer #1: Yes

4. Is the manuscript presented in an intelligible fashion and written in standard English?

Reviewer #1: Yes

5. Review Comments to the Author

Reviewer #1: Reviewer comments

The article is well written, experimental work is adecuate and conclusions are correct.

However, few errors or tipos are present in the text, such as:

Line 158, there are two spaces among genes pooled

Line 178, Fig. 2ABC should be Fig. 2A-C

Line 389, the two initial sentences in conclusion ate not conclusions of this work, please delete

Line 405, 450 space mM, 10 space mM and all over M&M

Line 413, 0.2 space μL

Line 432, change V to v for volumen

Line 434, change EtOH to ethanol

Line 445, change NaOAc to ammonium acetate

Line 455 change 10 ng/uL to 10 ng/μL

Revise reference format , there are several tipos

Ref 8 in the number of the article

Ref 13 Franco P-O should be Franco PO, and Le Bail A

Ref 15 Extremophiles. Delete colon

Ref 36 is a journal bioRXIV?

Ref52 delete capital letters in the title as well as in ref 53 and 54

Ref 62. Delete MGG before Mol, Gen Genet

6. PLOS authors have the option to publish the peer review history of their article (what does this mean?). If published, this will include your full peer review and any attached files.

Reviewer #1: No

---

## [Author Response · Author response to Decision Letter 0]

25 Aug 2020

As stated in the Response to Reviewers Document, copied here:

We thank the editor and reviewer for their careful reading of our manuscript. We are pleased to provide a corrected manuscript, with the details of changes outlined below.

RESPONSE TO REVIEWERS

Editor comments:

1. Parameters applied for RankAggreg (CE Kendall or CE Spearman? Weight settings? What stopping criteria?) to ensure replicability. 

RESPONSE:

Added parameters used in Materials and Methods as follows at line 497:

‘In addition, a rank aggregation method based on a Monte Carlo cross-entropy algorithm (R-package; RankAggreg 0.6.5; https://CRAN.R-project.org/package=RankAggreg; with default settings and following set parameters: method= "CE", distance= ”Spearman”, convIn= 7, seed=100) was used to combine the gene ranks of the three above algorithms and create a consensus housekeeping gene ranking.’

2. A MIQE checklist (https://rdml.org/miqe.html) should be added as supplement information and experimental details supplied accordingly.

RESPONSE:

Added to the Supplement as Table S2.

Reviewer comments:

in the text, such as:

Line 158, there are two spaces among genes pooled

Line 178, Fig. 2ABC should be Fig. 2A-C

Line 389, the two initial sentences in conclusion are not conclusions of this work, please delete

Line 405, 450 space mM, 10 space mM and all over M&M

Line 413, 0.2 space μL

Line 432, change V to v for volumen

Line 434, change EtOH to ethanol

Line 445, change NaOAc to ammonium acetate

Line 455 change 10 ng/uL to 10 ng/μL

Revise reference format , there are several tipos

Ref 8 in the number of the article

Ref 13 Franco P-O should be Franco PO, and Le Bail A

Ref 15 Extremophiles. Delete colon

Ref 36 is a journal bioRXIV?

Ref52 delete capital letters in the title as well as in ref 53 and 54

Ref 62. Delete MGG before Mol, Gen Genet

RESPONSE:

Line 389 – removed first sentence, but kept second sentence due to better introduction to our conclusions.

Ref 36 – BioRxiv is a preprint server: 

(from PLOS Submission guidelines: “Any and all available works can be cited in the reference list. Acceptable sources include:

- Published or accepted manuscripts

- Manuscripts on preprint servers, providing the manuscript has a citable DOI or arXiv URL.”)

Corrected all typos and errors.

---

## [Editor Report · Decision Letter 1]

19 Feb 2021

Identification and selection of optimal reference genes for qPCR-based gene expression analysis in Fucus distichus under various abiotic stresses

PONE-D-20-12656R1

Dear Dr. Braybrook,

We’re pleased to inform you that your manuscript has been judged scientifically suitable for publication and will be formally accepted for publication once it meets all outstanding technical requirements.

Kind regards,

Christian Schönbach, Dr.rer.nat.

Section Editor

PLOS ONE

Additional Editor Comments (optional):

All comments were satisfactorily addressed.
---

## [Editor Report · Acceptance letter]

16 Apr 2021

PONE-D-20-12656R1 

**Identification and selection of optimal reference genes for qPCR-based gene expression analysis in *Fucus distichus* under various abiotic stresses**

Dear Dr. Braybrook:

I'm pleased to inform you that your manuscript has been deemed suitable for publication in PLOS ONE. Congratulations! Your manuscript is now with our production department. 

Kind regards, 

on behalf of

Dr. Christian Schönbach 

Section Editor

PLOS ONE